# Gene Silencing through CRISPR Interference in Mycoplasmas

**DOI:** 10.3390/microorganisms10061159

**Published:** 2022-06-05

**Authors:** Daria V. Evsyutina, Gleb Y. Fisunov, Olga V. Pobeguts, Sergey I. Kovalchuk, Vadim M. Govorun

**Affiliations:** 1Scientific Research Institute for Systems Biology and Medicine, 117246 Moscow, Russia; herr.romanoff@gmail.com (G.Y.F.); vgovorun@yandex.ru (V.M.G.); 2Federal Research and Clinical Center of Physical-Chemical Medicine, 119435 Moscow, Russia; nikitishena@mail.ru; 3Shemyakin-Ovchinnikov Institute of Bioorganic Chemistry, 117997 Moscow, Russia; xerx222@gmail.com

**Keywords:** CRISPRi, mycoplasma, transcription factors, knockdown

## Abstract

Mycoplasmas are pathogenic, genome-reduced bacteria. The development of such fields of science as system and synthetic biology is closely associated with them. Despite intensive research of different representatives of this genus, genetic manipulations remain challenging in mycoplasmas. Here we demonstrate a single-plasmid transposon-based CRISPRi system for the repression of gene expression in mycoplasmas. We show that selected expression determinants provide a level of dCas9 that does not lead to a significant slow-down of mycoplasma growth. For the first time we describe the proteomic response of genome-reduced bacteria to the expression of exogenous *dcas9*. The functionality of the resulting vector is confirmed by targeting the three genes coding transcription factors-*fur*, essential *spxA*, *whiA*, and histone-like protein *hup1* in *Mycoplasma gallisepticum*. As a result, the expression level of each gene was decreased tenfold and influenced the mRNA level of predicted targets of transcription factors. To illustrate the versatility of this vector, we performed a knockdown of metabolic genes in a representative member of another cluster of the Mycoplasma genus-*Mycoplasma hominis*. The developed CRISPRi system is a powerful tool to discover the functioning of genes that are essential, decipher regulatory networks and that can help to identify novel drug targets to control Mycoplasma infections.

## 1. Introduction

Mycoplasmas are genome-reduced wall-less bacteria. The evolution of this group that is capable of infecting a wide range of hosts led to the reduction of metabolic pathways [1]. Despite this, most species can be grown in the laboratory in a rich medium supplemented with species-specific energy sources like arginine or urea [2]. These bacteria demonstrate a broad adaptive capability in perturbation model experiments. Moreover, in nature, mycoplasmas are successful pathogens [3]. The development of such a field of science as synthetic biology is closely associated with mycoplasmas. In an artificial minimal bacterium, *Mycoplasma mycoides* JCVI-syn3A with 452 protein-coding genes, about one third has no known function [4]. Many of these genes are often unique to mycoplasmas. At the same time, genes kept in genome-reduced bacteria can either have broader functions like Eno [5,6], GAPDH [7], EF-Tu [8] or represent a new alternative group of genes that are involved in vital processes. A high proportion of essential genes is shown for mycoplasmas [9,10]. Elucidating the role of essential genes, as well as genes with unknown functions, is crucial for understanding how bacterial growth is controlled, the synchronization of cellular processes, and the discovery of the functions of new, previously uncharacterized genes. A barrier in the study of such unique bacteria as mycoplasmas is the absence of universal genetic manipulation tools. It was demonstrated that the transposon-based vectors could be successfully used for random insertion [11,12]. Such systems can be used for the random inactivation of genes. In addition, this method allows for the introduction of the target genes into the genome of mycoplasmas as part of a transposon and can be used for gene overexpression [13,14]. However, this method is not suitable for figuring out the exact function of essential genes. There have been attempts to develop a self-replicating plasmid for some mycoplasma species [15,16]. Such systems simplify the work, since they do not require mapping the location of the transposon insertion into the genome and exclude the influence of the insertion position in the genome, but they are not universal. It was shown that oriC, a necessary determinant of plasmid replication, was very specific to their host species and could not be shared even for closely related mycoplasma species [15]. Recently, to investigate the function of essential genes, CRISPR interference (CRISPRi) has been developed [17]. This emerging technology exploits the catalytically inactive Cas9 (dCas9) and single guide RNA (sgRNA) to repress sequence-specific genes [17]. The first and only study describing the successful gene silencing through CRISPR interference in *Mycoplasma pneumonia* has been reported [18]. In addition, the successful use of the endogenous CRISPR/Cas system in *Mycoplasma gallisepticum* to edit genes was reported [19,20]. The disadvantage of using the endogenous system is the limitation in further modification of the system. An example would be the design of inducible tools for genetic manipulation. Moreover, some mycoplasma species including human pathogenic bacteria infecting pulmonary and urogenital tracts like *Mycoplasma pneumonia* and *Mycoplasma hominis* lacked the CRISPR/Cas system [21]. In this study we developed transposon-based plasmids for regulating the knockdown of gene expression in mycoplasmas through CRISPRi. We show that our proposed system provides for the reliable expression dCas9 of *Streptococcus pyogenes* (*S. pyogenes*) that does not lead to a significant delay in the growth rate. We also evaluated the effect of dCas9 expression on protein levels in *Mycoplasma gallisepticum*. Using the developed CRISPRi system, we performed knock-down of three genes coding transcription factors (TFs): *fur*, essential *spxA*, *whiA*, and *hup1*. The latter was selected since it has been demonstrated that HU-1 of *M. gallisepticum* lacks non-specific DNA binding activity [22] and thus may have evolved towards site-specific DNA binding. We also demonstrate that TFs repression influences the expression of some of their predicted targets. Our system was tested on *M. gallisepticum* and *M. hominis* and thus is universal for mycoplasmas.

## 2. Materials and Methods

### 2.1. Bacterial Strains and Growth Conditions

*Mycoplasma gallisepticum S6* and mutant strains were cultivated on a liquid medium containing tryptose (20 g/L), Tris (3 g/L), NaCl (5 g/L), KCl (5 g/L), yeast extract (1%), horse serum (10%) and glucose (1%) at pH 7.4 and 37 °C in aerobic conditions. For monitoring the growth rate of *M. gallsepticum S6* and *dcas9*-transformants, bacteria were grown in a liquid medium supplemented with 0.002% phenol red. The ratio of absorbance at 430 nm and 560 nm of the culture medium [23] was measured each 30 min for 28 h and growth curves were created. Bacteria-free media was used as a negative control. All measurements were performed in quintuple. The initial cell biomass calculated as the amount of genomic DNA was equal for all bacteria cultures.

*Mycoplasma hominis H34* and mutant strains were grown on a brain heart infusion medium (DIFCO, Franklin Lakes, NJ, USA) supplemented with 10% horse serum (Biolot, Moscow, Russia), 1% yeast extract (Helicon, Moscow, Russia), 1% arginine, and penicillin (Sintez, Saint-Petersburg, Russia), with a final concentration of 500 units/mL. The culture was grown at 37 °C in aerobic conditions.

### 2.2. Plasmid Construction

To generate the pRM5L2-dcas9 plasmid for delivery and expression of dCas9 *S. pyogenes* in mycoplasma the following was done: the wild-type cas9 *S. pyogenes* was mutated through site-directed mutagenesis in two steps. In the first step, two PCR reactions were performed to change the Asp-10 codon to Ala (GAT → GCT), His-840 codon to Ala (CAC → GCT) and also TGA to TAA stop-codon. For this, wild-type cas9 was amplified using Phusion High-Fidelity DNA Polymerase (Thermo Fisher Scientific, Waltham, MA, USA) with two primer pairs (dCas9_mut_1_F, dCas9_mut_1_R) and (dCas9_mut_2_F, dCas9_mut_2_R). In the second step, the obtained amplicons were merged via PCR to generate a full-length dcas9 gene. Primers dCas9_mut_1a_F with a non-complementary 5′-end and dCas9_mut_2_R were used. All of the oligonucleotides used for PCR are listed in Appendix A. The resulting product was then purified by chloroform extraction and isopropanol precipitation, and cloned into XhoI- cut pRM5L2 vector using NEBuilder HiFi Assembly Master Mix (NEB). The product was transformed into TOP10 chemically competent *E. coli* (Thermo Fisher Scientific, Waltham, MA, USA) and plated onto Luria–Bertani (LB)–ampicillin. Colonies were picked, screened for insertion, enriched in LB-ampicillin media and partially sequenced. The plasmid containing the desired mutations in the cas9 was called-pRM5L2-dcas9 and used as a precursor in further experiments.

The fragments of dsDNA coding sgRNAs were assembled from chemically synthesized oligonucleotides (Appendix A). Each fragment consisted of seven overlapping oligonucleotides, four of which were common to all sgRNAs-coding fragments. For the fragments assembly into dsDNA, 100 pmol of each of seven oligonucleotides were 5’-phosphorylated with T4 Polynucleotide Kinase (Thermo Fisher Scientific, Waltham, MA, USA) in 20 μL reaction volume. Next, 5 μL of each oligos were mixed, annealed (incubated at 98 °C for 2 min, followed by cooling to 25 °C at 0.1 °C/c) and ligated using T4 DNA ligase (Thermo Fisher Scientific, Waltham, MA, USA) into 40 μL. The resulting product was purified by chloroform extraction and isopropanol precipitation, and ligated with T4 DNA Ligase to pRM5L2-dcas9 vector between ApaI, XbaI sites. The standard procedures for obtaining a vector were done as described above.

### 2.3. Transformation

All vectors used for transformation of *M. gallisepticum* and *M. hominis* were based on the pRM5L2 transposon vector used in previous work [14] and listed in Appendix A. Transformation of mycoplasmas was performed as described earlier [24]. Briefly,1 mL of cells at the exponential growth phase were harvested by centrifugation at 4 °C at 8000 g (for *Mycoplasma gallisepticum*) or 10,000 g (for *M. hominis*) for 10 min. The pellet was resuspended in 250 μL of electroporation buffer (8 mM HEPES, 272 mM sucrose, pH = 7.4). The procedure was repeated twice to remove traces of the growth medium. The cell suspension and plasmid (400 or 800 ng) were then transferred to the electroporation cuvette and the pulse was performed using a Gene Pulser device (Bio-Rad, Hercules, CA, USA). The parameters for the cuvette and the pulse are: 2 mm cuvette, voltage 2500 V, capacity 100 Ω, resistance 25 μF. The cells then were transferred to 1 mL of fresh appropriate medium and grown for 4 or 8 h, for *M. gallisepticum* or *M. hominis*, respectively. The cells were then cultivated in a semi-liquid appropriate medium (identical to liquid growth medium) containing 0.3% agar and 2 μg/mL of tetracycline until the visible colonies were formed. The first colonies of *M. gallisepticum* appeared on the fifth day of growth, and colonies of *M. hominis* appeared on the eighth day. For transformants, the time of colony appearance and its size can be an indirect indicator of the influence of transposon position to bacteria fitness. In our study, we collected early-appearing big colonies. The colonies were picked and grown in liquid medium supplied with 2 μg/mL of tetracycline. Two transformants for each plasmid were used in further experiments.

### 2.4. Western Blot Analysis

Twenty mL of the cultures were harvested by centrifugation (10,000 g for 10 min at 4 °C) and the cell pellets were washed twice with a Tris buffer (50 mM Tris-HCl, pH 7.4, 150 mM NaCl, and 3 mM MgCl_2_). Subsequently, the cell pellets were resuspended in a 200 μL Tris buffer. The cells were lysed by adding 5 μL of 10% NP-40 (Sigma-Aldrich, St. Louis, MO, USA) solution. Lysates were frozen at −75 °C for at least 1 h thawed. One μL of protease inhibitor cocktail (GE HealthCare, Chicago, IL, USA) and Nuclease Mix (GE HealthCare, Chicago, IL, USA) were added and samples were incubated for 30 min at 22 °C. Cell debris was removed by centrifugation (16,000× *g* for 30 min at 4 °C). Protein concentration in the supernatant (cell free extracts) was determined using the Bradford protein assays (Bio-Rad, Hercules, CA, USA) and bovine serum albumin as reference. Cell free extracts containing 20 μg of total protein were loaded on 6–12% gradient polyacrylamide gels, and 5 μL of Precision Plus Protein™ WesternC™ Blotting Standards (Bio-Rad, Hercules, CA, USA) or PageRuler™ Plus Prestained Protein Ladder (Thermo Fisher Scientific, Waltham, MA, USA) were used as a molecular weight ladder. After electrophoresis, the proteins were transferred by semi-dry electroblotting using a Trans-Blot Turbo Transfer System (Bio-Rad, Hercules, CA, USA) onto a polyvinylidene fluoride (PVDF) membrane (Bio-Rad, Hercules, CA, USA) at 2.5 A and 1 V for 30 min. The membrane was washed with Tris buffered saline (TBS), incubated at 22 °C in TBST buffer (TBS containing 0.05% (*v*/*v*) Tween 20) with 3% Albumin (BSA) Fraction (Panreac, Spain) for 1 h and then washed 2 times for 5 min each with TBST. The PDVF membrane was incubated for 1 h at 22 °C with primary antibody solution (Cas9 Monoclonal Antibody 10C11-A12, Invitrogen, Waltham, MA, USA); diluted 1:1000 in TBST. Afterwards, the membrane was washed three times for 10 min each with TBST and incubated with the secondary antibody (Anti-Mouse IgG (Fc specific)-Peroxidase, Cat. # A2554, Sigma-Aldrich, St. Louis, MO, USA), diluted 1:100000 in TBST for 1 h at 22 °C, followed by three times washing with TBST for 10 min. The blots were developed by the chemiluminescence method with Amersham™ ECL Western Blotting Detection Reagents (Little Chalfont, UK). The protein weight markers were visualized in fluorescence mode. The blot images were acquired through the ChemiDoc MP imaging System (Bio-Rad, Hercules, CA, USA). Semiquantification of the bands was carried out by optical densitometry and analyzed using the Image Lab Software (Bio-Rad, Hercules, CA, USA).

### 2.5. RNA Purification and cDNA Synthesis

RNA was isolated as previously described [25]. One-hundred-microliter aliquots of mycoplasma culture were directly lysed in TRIzol LS reagent (Life Technologies, Carlsbad, CA, USA) at a 1:3 ratio of culture medium: TRIzol LS (*v*/*v*). The nucleic acids were extracted with chloroform and precipitated by the addition of an equal volume of isopropanol followed by centrifugation. The pellets were washed with 80% ethanol and finally resuspended in 20 μL of mQ (Panreac, Barcelona, Spain). The amount of RNA was determined using a Qubit 2.0 fluorometer (Thermo Fisher Scientific, Waltham, MA, USA). The resulting RNA was treated with DNAse I (Thermo Fisher Scientific, Waltham, MA, USA), and cDNA was synthesized from random hexamer primers by using Maxima H Minus Reverse Transcriptase (Thermo Fisher Scientific, Waltham, MA, USA), according to the manufacturer’s protocol.

### 2.6. Quantitative RT-PCR

Quantitative real-time PCR was performed using dNTP, PCR buffer, Taq-polymerase (Lytech, Russia), SYBR Green I (Invitrogen, Waltham, MA, USA), and CFX96TM Real-Time PCR Detection System (Bio-Rad, Hercules, CA, USA). The primers used are listed in Appendix A. All primers were designed using a BAC Browser [26]. Each 20-μL reaction contained 0.2 μL of template cDNA. The thermal cycling conditions were as follows: initial denaturation at 95 °C for 1 min; then 40-cycle amplification (94 °C for 15 s, 58 °C for 20 s, and 68 °C for 1 min). The melting curve was obtained by gradually heating the PCR mixture from 65–94 °C at a rate of 0.5 °C every 5 s, with continuous fluorescence scanning. The relative expression for each sample was determined using the 2^−ΔΔCt^ method and normalized to the amount of *tuf* transcripts present in the RNA samples. qRT-PCR experiments were performed on three replicates per each transformant. The significance of the difference between the two mean values was assessed using the Wilcoxon rank test implemented in the R software package. *p* < 0.05 (*); ns, no significant difference.

### 2.7. Proteomic Analysis

Sample preparation for proteomic analysis was performed as described in [27]. Briefly, samples (10 mL of log-phase growing cells for each sample) were lysed in a buffer containing 1% sodium deoxycholate (DCNa) (Sigma-Aldrich, St. Louis, MO, USA) and 100 mM Tris-HCl (pH 8.5) with a protease inhibitor cocktail (GE Healthcare, Chicago, IL, USA) through ultrasonication with a Branson 1510 sonicator at 4 °C for 1 min. Protein concentration was estimated using the BCA Assay Kit (Sigma-Aldrich, St. Louis, MO, USA). Disulfide bonds were reduced in supernatant (containing 300 μg of total protein) by the addition of Tris(2-carboxyethyl)phosphine hydrochloride (TCEP) (Sigma-Aldrich, St. Louis, MO, USA) to a final concentration of 5 mM and reaction was incubated for 60 min at 37 °C. To alkylate free cysteines, chloroacetamide (Sigma-Aldrich, St. Louis, MO, USA) was added to a final concentration of 30 mM and placed at room temperature in the dark for 30 min. Trypsin Gold (Promega, Madison, WI, USA) was added for a final trypsin:protein ratio of 1:100 (*w*/*w*) and incubated at 37 °C overnight. To stop trypsinolysis and degrade the acid-labile DCNa, trifluoroacetic acid (TFA) was added to the final concentration of 0.5% (*v*/*v*), incubated at 37 °C for 45 min and the samples were centrifuged at 14,000× *g* for 10 min to remove the DCNa. Peptide extract was desalted using a OASIS columns (Waters, Milford, MA, USA) according to the manufacturer’s protocol and analyzed by liquid chromatography-mass spectrometry (LC-MS). LC-MS analysis was carried out on an Ultimate 3000 RSLC nano HPLC system connected to a QExactive Plus mass spectrometer (Thermo Fisher Scientific, Waltham, MA, USA). Samples were loaded to a home-made trap column 20 mm × 0.1 mm, packed with Inertsil ODS3 3 μm sorbent (GL Sciences, Tokyo, Japan), in the loading buffer (2% ACN, 98% H_2_O, 0.1% TFA) at 10 μL/min flow and separated at RT in a home-packed fused-silica column 500 × 0.1 mm packed with Reprosil PUR C18AQ 1.9 (Dr. Maisch, Beim Brückle, Germany) into the emitter prepared with P2000 Laser Puller (Sutter, Novato, CA, USA) [28]. Samples were eluted with a linear gradient of 80% ACN, 19.9% H_2_O, 0.1% FA (buffer B) in 99.9% H_2_O, 0.1% FA (solvent A) from 4 to 36% of solvent B in 1 h at 0.44 μL/min flow at 20 °C. MS data were collected in DDA mode. Data are available via PRIDE database, project ID PXD026928 (http://www.ebi.ac.uk/pride/archive/projects/PXD026928 (accessed on 26 June 2021)).

Raw files for wild-type *M. gallisepticum* and *dcas9*-expressing transformants were extracted and searched against the database, which contained the *M. gallisepticum S6* Uniprot reference database, protein sequences of dCas9 from *S. pyogenes* and TetM, using MaxQuant v 1.6.6.0 Software with the default settings. The proteinsgroups.txt file containing LFQ intensities based on the MS level peak areas was loaded into R and analyzed for differential expression with the DEP package [29]. Data was filtered to exclude proteins that were not identified in two out of three replicates of at least one condition. Missing values were then imputed using random draws from a Gaussian distribution centered around a minimal value. Differential abundance analyses of proteins in dcas9-expression transformants versus wild-type *M. gallisepticum S6* (Mga) were performed using the empirical Bayes method available in DEP (uses limma). The resulting *p*-values were adjusted using the Benjamini-Hochberg approach and the significance threshold was set at an adjusted *p*-value of 0.05 and a log2(fold change) of 1. Results of the differential abundance analyses are available in Appendix A.

### 2.8. In Silico Analysis

The CUSP program of EMBOSS was used to create a codon usage table for *Mycoplasma gallisepticum S6*, *Mycoplasma hominis H-34* and *cas9* sequence from *S. pyogenes*.

For sgRNAs secondary structure prediction, The ViennaRNA Web Services (http://rna.tbi.univie.ac.at/) were applied. The DNA targeting specificity of designed sgRNAs was verified using blastn (https://blast.ncbi.nlm.nih.gov/Blast.cgi).

## 3. Results and Discussion

### 3.1. Construction of a CRISPRi System for Silencing Gene Expression in Mycoplasma

CRISPRi is based on the ability of dead Cas9 protein in complex with sgRNA to specifically bind to the target gene and sterically hinder transcription at the sgRNA base-pairing genomic locus. One of the most commonly used Cas9 proteins is derived from *S. pyogenes*. It recognizes short DNA sequence NGG as a protospacer adjacent motif (PAM). *S. pyogenes* is a Gram-positive pathogen, and the genome consists of a single circular chromosome with an average GC content of 38.5% [30]. The high AT content imposed an unusual codon bias can be a problem for the expression of *cas9* in other bacteria or mammalian cells. However, for mycoplasmas, which are characterized by a low GC content (31% for *Mycoplasma gallisepticum* and 27% for *Mycoplasma hominis*) it becomes an advantage. The only exception is a specific usage of the UGA codon to encode tryptophan in mycoplasmas. In other bacteria including *S. pyogenes*, UGA (TGA) is a stop codon. To generate the inactive dCas9 from wild-type *cas9 S. pyogenes*, we change Asp-10 and His-840 codons to alanyne codons (GAT → GCT and CAC → GCT, respectively) [17] and TGA was also replaced with a TAA stop codon through site-directed mutagenesis (SDM) (Figure 1A).

The pRLM5L2-dcas9 is a derived from pRLM5L2, which is a 64575 bp Tet^R^ transposon-based vector [24]. The pRLM5L2 plasmid contains XhoI and NcoI sites downstream to mycoplasma strong constitutive promoters and ribosome-binding sites (RBS) for cloning and expression of *dcas9* (Figure 1B and Appendix A). However, *dcas9* contains an internal NcoI site, so we decided to clone *dcas9* into the XhoI-cut pRM5L2 vector. In order to ensure the optimal length (8 nt) of the spacer between the Shine–Dalgarno (SD) box and AUG start codon, we used a primer (dCas9_mut_1a_F) with a non-complementary 5′-end. The selected sequences of the promoter and SD-box were tested in reporter constructs in our previous study [5]. The resulting plasmid was annotated as pRLM5L2-dcas9 (Figure 1C). We transformed this vector into *M. gallisepticum S6* and *M. hominis H-34*. The average transformation efficiencies (in number of transformants/CFU/μg of plasmid) for *M. gallisepticum* and *M. hominis* were 6.2 × 10^−8^ and 3.5 × 10^−7^, respectively.

### 3.2. pRLM5L2-dcas9 Provides dcas9 Expression at Both mRNA and Protein Levels in Two Mycoplasma Species

We selected several transformants based on the time of colony appearance and further on the position of transposon integration into the genome. The *dcas9* expression was confirmed at both mRNA and protein levels using real-time quantitative PCR and western blotting for three transformants of *M. gallisepticum* and two transformants for *M. hominis* (Figure 2A–C).

The expression level of *dcas9* normalized to the *tuf* level was equal in both species (on average 0.43 and 0.38, respectively).

### 3.3. The Selected Expression Level of dCas9 Does Not Provide a Toxic Effect for M. gallisepticum

It is known that high-level expression of dCas9 may cause severe growth inhibition, by up to approximately 50% compared to that in wild-type *E. coli* [31]. To test if the resulting expression level of dCas9 alone affects the growth of *M. gallisepticum*, we evaluated its growth curves by determining the growth index value [23], which is the ratio of absorbance at 430 nm and 560 nm of the culture medium (Figure 3A).

We added the pH indicator phenol red in the culture medium. *M. gallisepticum* produces energy through glycolysis and regenerates NAD^+^ by the reduction of pyruvate to lactate, which leads to a decline in the pH medium. The carrying capacity for wild-type *M. gallisepticum* (Mga) and two *dcas9*-expression transformants (dCas9-1 and dCas9-3) was 2.24, 2.21 and 2.17. The carrying capacity is the maximum number, density, or biomass of a population that an environment can sustain [32]. Here, it is a maximum value of the growth index. The time when bacteria reached half of the carrying capacity was ~14 h for Mga, ~15 h for dCas9_1 and ~15.5 h for dCas9_3. Thus, we conclude that the observed expression level of dCas9 does not lead to significant growth slowdown of mycoplasma.

We next explored changes in the protein level to figure out the effect of dCas9 on the physiology of *Mycoplasma gallisepticum*. For this, we performed a quantitative proteomic analysis of two dCas9-expressing transformants of *M. gallisepticum* (dCas9-1, dCas9-3) and wild-type *M. gallisepticum* (Mga). The abundance of 23 proteins for dCas9-1 and 16 for dCas9-3 was significantly changed with |log2 (fold change)| > 1 under dCas9-overexpression compared to the protein level in wild-type *M. gallisepticum* (Figure 3B). Among those, the change in the level of seven proteins was co-directed in both strains (four were up-regulated, three were down-regulated). The independent analysis of two dCas9-expressing transformants allowed us to make selections specific to every strain and common for both strains’ protein changes. As expected, protein levels of dCas9 and selective marker TetM were high for both transformants (Figure 3C). Wild-type Mycoplasma does not express dCas9 and TetM. To calculate the change in the level of these proteins, the missing values were imputed as described in section “Material and Methods”. Other upregulated proteins in both transformants are AtpD_2 and uncharacterized protein GCW_03660. Genes encoding these proteins with another five genes are composed of an operon [24]. AtpD_2 is related to the beta subunit of type 3 F_1_F_0_ ATPase [33] and its role has been suggested to promote substrate capture in the immunoglobulin binding and proteolysis (MIP-MIB system) [34]. The levels of amino acid permease (GCW_01790), Bmp domain-containing protein (GCW_03380), and S8 family serine peptidase (GCW_00525) were decreased while dCas9 was expressed. The exact role of these proteins is unknown, but they can be part of amino acids and protein turnover [35,36] involved in nutrition or virulence [37,38]. About half of the proteins whose abundance has changed belong to variable lipoprotein and hemagglutinin VlhA. In *M. gallisepticum*, the variable lipoprotein and hemagglutinin (*vlhA*) gene family consists of over 40 closely related genes. It was shown that *M. gallisepticum* expressed only one *vlhA* family member at a time at a high level, while the expression of others was minor. Changes in the expression pattern were observed during infection progression and differed between strains [14,39,40]. In *dcas9*-expressing transformants, the abundance of major VlhA (GCW_01940) changed two or five-fold. During infection in vivo, it can be changed from a factor of ten to a hundred times. We supposed that VlhA protein change in dCas9 transformants could be explained by cell-to-cell variation. For *E. coli* it was shown that dCas9 expression led to abnormal cell division resulting in a linear filamentous phenotype [31]. In our previous work we observed a similar phenotype of *M. gallisepticum* when the transcription regulator MraZ was overexpressed [13]. Remarkably, in both dCas9-expression transformants of Mycoplasma, the level of MraZ was significantly reduced, but only 1.5 and 1.8-fold. This may affect the morphology of cells. These proteomic changes can be induced by the non-specific interaction between dCas9 or TetM and nucleic acids, endogenous proteins, or may be a stress response associated with the heterologous expression or both.

### 3.4. Design of sgRNA for Rapid Assembly and Cloning into pRLM5L2-dcas9

A single-guide RNA is a necessary component of CRISPRi, which together with protein dCas9, targets the complex to a specific DNA region. Each sgRNAs consists of three segments: a 20-nucleotide(nt) target-specific complementary region, a 42 nt Cas9-binding hairpin (Cas9 handle), and a 40 nt transcription terminator derived from *S. pyogenes* [17]. pRLM5L2-*dcas9* contains ApaI and XbaI sites with a downstream AT-rich terminator (AT ~40%) that differs from the canonical *S. pyogenes* sequence (AT ~29%). In initial studies, we tried to generate sgRNA using the terminator sequence provided in the plasmid. The predicted secondary structure of designed sgRNA1 for *fur* exhibits a significant level of base pair interactions between the 20-nt target sequence and other RNA sites, compromising the formation of essential stem-loop structures (Appendix A). This sgRNA was not effective for the repression of *fur*, instead sgRNA that contained the same 20-nt target sequence and canonical terminator from *S. pyogenes* provided 22.5-fold *fur*-downregulation (Appendix A). We hypothesized that the high AT content of the terminator together with AT-rich targets increases the probability of intramolecular interactions in sgRNA. In further work, we used only the standard Spy terminator sequence.

The fragments of dsDNA coding sgRNAs were assembled from chemically synthesized oligonucleotides (Appendix A). Each fragment consisted of seven overlapping oligonucleotides, four of which were common to all sgRNA-coding fragments (Figure 1D). We used a strong constitutive promoter for sgRNA expression. It is helpful to tune gene repression because some essential genes are sensitive to dramatic knockdown. This can be fulfilled by titrating the concentration of dCas9 or sgRNA. In mycoplasmas, as in other bacteria, a change in the promoter elements, such as −10 box, EXT element, initiator nucleotide or spacers between them can provide a hundredfold shift in the RNA level [24]. In our construction, the promoter sequence for sgRNA expression is included in two of the three target-specific oligonucleotides (sg3 and sg4). Replacement of these oligos would allow the sgRNA expression to vary.

### 3.5. Application of CRISPRi for Gene Knockdown in Two Mycoplasma Species

To demonstrate the feasibility of the CRISPRi system in *M. gallisepticum*, we engineered the sgRNAs for the repression of *hup1*, *fur*, *whiA* and the essential gene *spxA* [41]. Fur, WhiA and SpxA are transcription regulators [42,43], while Hup1 is one of the two histone-like proteins [25]. In our previous studies, we demonstrated that the transposon-based vector pRM5L2 originally developed for *M. gallisepticum* [13,24] could be used for other representatives of mollicutes, like *M. hominis* [44]. Therefore, we applied the designed CRISPRi system for the knockdown *deoA* which codes thymidine phosphorylase in *M. hominis*. We generated sgRNAs (Table 1) that target the nontemplate strand of genes. The location of targets was adjacent to a protospacer motif (PAM) for dCas9 *S. pyogenes*. For all target-genes two sgRNAs were used; all of them were complementary to the coding region (Appendix A). Due to the high AT content of mycoplasmal genomes, the choice of the targeted region with downstream PAM 5’-NGG-3’ is restricted. On the other hand, this factor can minimize off target effects. Using qRT-PCR, we measured the level of mRNA of all targeted genes in mycoplasma transformants and compared them to expression level in wild-type Mycoplasma (Figure 4).

The maximum value of repression was more than 40-fold with fur-sg2 in *M. gallisepticum* and deoA-sg2 in *M. hominis*. The expression level of *eno* encoding enolase was not changed for most transformants (Appendix A). We have analyzed the thermodynamic properties of the designed sgRNAs sequences and we did not find any dependence of sgRNAs activity on the GC content, but GU dinucleotide in the 3′ of targeting region of sgRNAs was associated with high repression of targets. It was demonstrated that silencing efficiency inversely correlated with the target distance from the translation start codon for *E. coli* [17]. Our results did not confirm this observation for mycoplasmas. Only *spxA*, sgRNA1 targeted to the nearest position to start-codon was more effective than sgRNA2. It means that other factors can determine CRISPRi silencing efficiency in genome-reduced bacteria with high AT content.

It has been shown that CRISPRi imposes a polar effect on upstream and downstream genes of the target gene in an operon [45]. According to our previous data, *fur* forms an operon with *nfo* and *GCW_02095* [24]. We compared the mRNA levels of these genes. The 43-foldrepresson of *fur* caused a 3.8-fold depletion of the co-transcribed downstream gene *GCW_02095* coding hypothetical protein DUF3196. In Peters’ study performed for *Bacillus subtilis* [45], all downstream genes in an operon showed equivalent knockdown. Such a difference from our data can be explained by the pronounced operon structure complexity in *M. gallisepticum*; on average, each coding operon has one internal transcriptional start site [24]. We did not observe the silence effect on upstreamed *nfo* (Figure 5A).

Next, we examined whether the repression of transcription factors via CRISPRi influenced the expression of their targets. Regulatory networks for WhiA and SpxA were determined in *Mycoplasma pneumoniae* [43]. We assumed that orthologous TFs kept the ability to regulate orthologous targets across one family of bacteria. This is a widely believed viewpoint that definitely has exceptions [46]. SpxA is an essential protein in *M. pneumoniae* that regulates itself and a regulon involved in the oxidative stress response (*msrA*, *osmC*; and *msrB*) [43]. In *M. gallisepticum, a* nine-fold repression of *spxA* expression by CRISPRi leads to a 2.6-fold downregulation of *osmC* and a slight decrease in the mRNA level of *msrB* and *osmC_2*, 1.7, and a 1.5-fold, respectively (Figure 5B). This low response can be explained by the absence of the oxidative condition that is necessary for processing disulfide bonds in protein SpxA [47]. WhiA represses one single ribosomal operon (mpn164-185, *rpsJ- adk*) [43], [Fisunov et al., to be published]. A 33-fold decrease in the expression level of *whiA* in *M. gallisepticum* expressing *dcas9* and sgRNA2 resulted in a 2.5-fold increased *rpsJ* level (Figure 5C). Thus, we conclude that gene knockdown in tenfolds via CRISPRi in *M. gallisepticum* allows us to observe regulation events at least at the mRNA level. For model bacteria like *E. coli* and *B. subtilis*, the range of gene repression implemented by CRISPRi is wider and is two to three orders of magnitude [17,45]. It remains an open question whether it is possible to achieve a comparable level of gene repression in genome-reduced bacteria with a meager repertoire of well-known regulators.

## 4. Conclusions

In the present study, a single-plasmid transposon-based CRISPRi system for the repression of gene expression in mycoplasmas was constructed. For the first time, the proteomic response of genome-reduced bacteria to the expression of exogenous *dcas9* was described. The functionality of the resulting vector was confirmed for two representative members of different clusters of the mycoplasma genus. This approach is useful in knocking down genes to better understand their function, and holds a great deal of promise to explore potential drug targets.

## Figures and Tables

**Figure 1 microorganisms-10-01159-f001:**
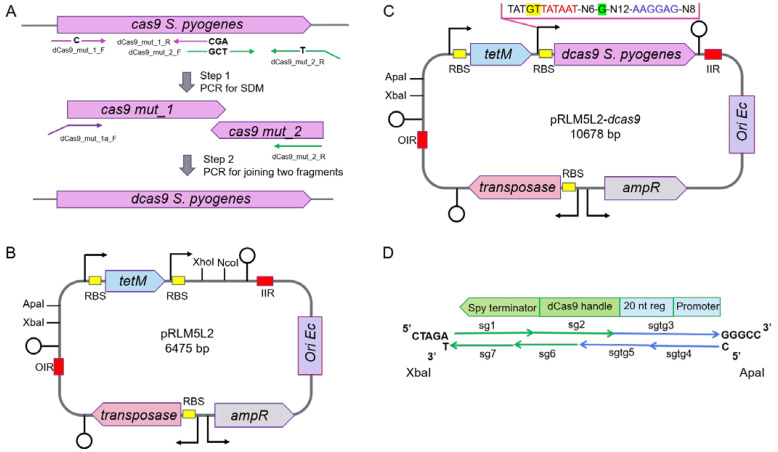
Schematic of the construction of components for CRISPRi (**A**) The *dcas9* gene was derived from wild-type *cas9 S. pyogenes* though site-directed mutagenesis (SDM) in two steps: firstly, we changed Asp-10 and His-840 codons to alanyne codons (GAT → GCT and CAC → GCT, respectively), and also TGA was replaced to TAA stop codon using PCR; in the second step, two fragments were linked to full-length *dcas9*. Primers are indicated by arrows; letters indicate substitute nucleotides. (**B**) pRLM5L2 transposon vector scheme. Promoters are shown as arrows, terminators–as circles, RBSs are yellow, OIR and IIR-inverted repeats for the integration transposon into the genome. (**C**) Vector map of the resulting pRM5L2-dcas9. Structure of promoter and RBS for *dcas9* expression in mycoplasmas are shown. The EXT element is highlighted in yellow, −10 box are red, initiator nucleotide is green, RBS is blue. (**D**) Strategy for assembly of dsDNA fragments coding sgRNAs. Each fragment consisted of seven overlapping oligonucleotides, four of which were common to all sgRNAs-coding fragments and three were unique for each specific sgRNA. All oligonucleotides were phosphorylated, ligated together and the resulting product was cloned between the ApaI and XbaI site in pRLM5L2-*dcas9*. Common oligonucleotides are green arrows, unique are blue. Sticky ends for cloning dsDNA fragments between ApaI and XbaI pre-cut pRLM5L2-*dcas9* are highlighted with letters. 20 nt reg is a base-pairing region of coding sgRNA. All oligonucleotides used in this study are listed in Appendix A.

**Figure 2 microorganisms-10-01159-f002:**
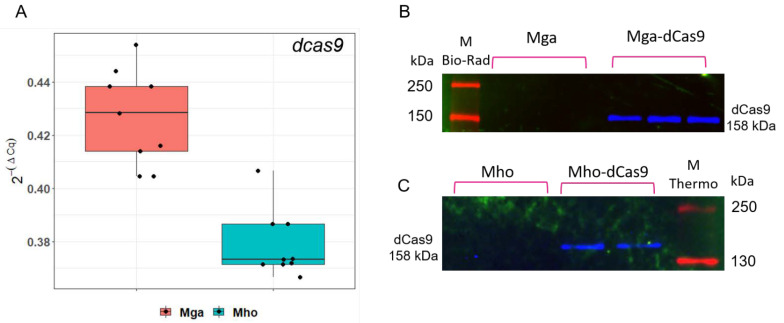
Validation of *dcas9* expression at mRNA and protein levels (**A**) Transcription analysis of *dcas9* by qRT-PCR in transformants of *M. gallisepticum* (Mga) and *M. hominis* (Mho). Quantitative *dcas9* expression was normalized to the expression level of *tuf* for both species. Boxplot was plotted from three repeated experiments (three independent transformants and three technical repeats for each). (**B**,**C**) dCas9 protein level by Western blot for wild-type *M. gallisepticum, M. hominis* and pRLM5L2-*dcas9* transformants.

**Figure 3 microorganisms-10-01159-f003:**
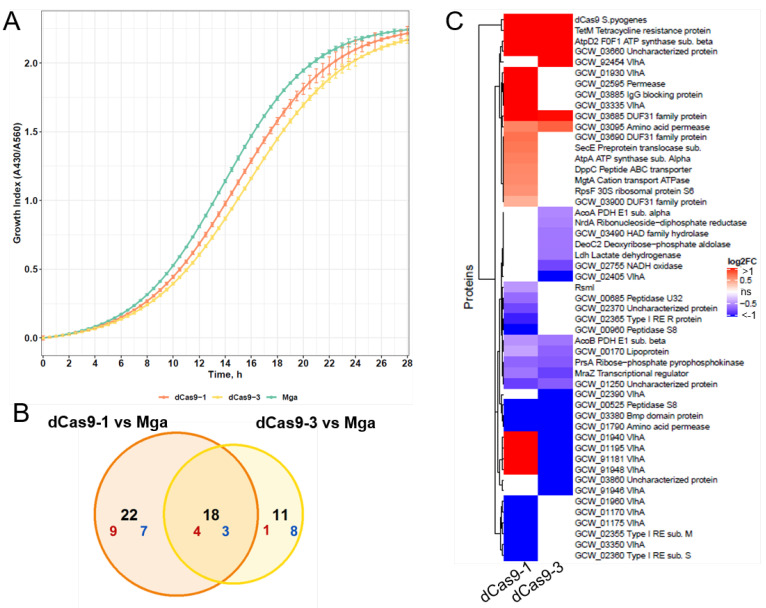
Effect of dCas9 expression on growth rate and protein levels in *M. gallisepticum* (**A**) Growth curves for wild-type *M. gallisepticum* (Mga) and two transformants expressed *cas9* (dCas9-1 and dCas9-3) were evaluated through the color change of the medium supplied with phenol red. (**B**) Venn diagram of the number of differentially expressed proteins from dCas9-1 and dCas9-3 transformants of *M. gallisepticum* when compared to wild type bacteria. The overlap shows the number of proteins that demonstrated co-directional change in two dCas9-expressed clones. The number of proteins with log fold change more than one are highlighted in red, with a log fold change less than one highlighted in blue. (**C**) The changes of protein levels while dCas9-expression is illustrated in the heatmap. White boxes are the proteins with non-significant changes (*p* adj. > 0.05).

**Figure 4 microorganisms-10-01159-f004:**
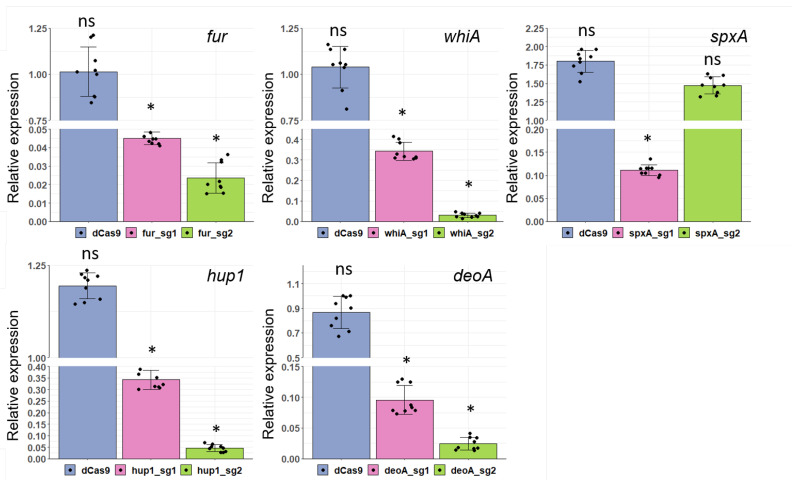
Efficiency of genes knockdown via CRISPRi in two mycoplasma species. Expression changes of target-genes (*fur*, *whiA*, *spxA* and *hup1* for *M. gallisepticum* and *deoA* for *M. hominis*) in transformants expressing dCas9 and corresponding sgRNAs compared to the wild-type mycoplasma. dCas9–transformants that express dCas9 and don’t express sgRNAs; sg1–constructed sgRNA1, sg2–constructed sgRNA2. The sequences of each designed sgRNAs and position of target sequences can be found in Table 1, Appendix A. (*: *p*-value < 0.05; ns: not significant).

**Figure 5 microorganisms-10-01159-f005:**
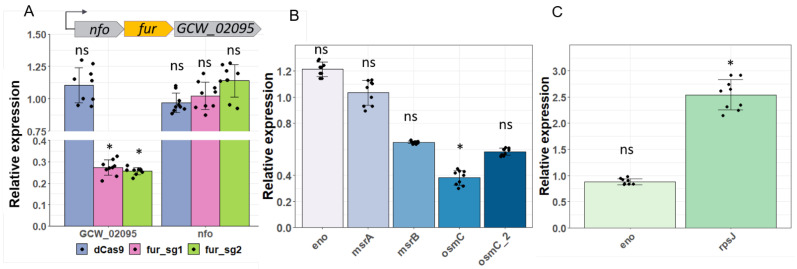
The effect of gene repression (**A**) Quantitative RT-PCR analysis showing levels of each gene of *fur*-containing operon in *dcas9* and *fur*-sgRNAs-expressing transformants compared to those in wild-type bacteria. Top panel is a scheme of operon structure. (**B**) Expression changes of SpxA-regulated genes in transformants expressing *dcas9* and the spxA-sgRNA1 (spxA-sg1) compared to *dcas9*-expressing transformants. As a control, expression change of *eno* is shown. (**C**) Expression level of *rpsJ*–target of transcription factor WhiA–in the simultaneous *dcas9* and whiA-sgRNA2 (whiA-sg2) expressing mycoplasma compared to *dcas9*-expressing transformants. (*: *p*-value < 0.05; ns: not significant).

**Table 1 microorganisms-10-01159-t001:** Characteristics of designed sgRNAs and their activity.

Name of sgRNA	20-nt Base-Pairing Region	Tm, °C	GC, %	Average Repression Range, Fold ± SE
fur-sg1	ACCTTCTAGGTTTGGTGATT	58.5	40	23 ± 0.6
fur-sg2	ACGAATCAACAAGATTGAGT	57.4	35	43 ± 8.8
whiA-sg1	CATTTTGCACCAACAAGTCG	60.9	45	3 ± 0.2
whiA-sg2	AGATGATCGTTTTTGGGTGT	59.9	40	33 ± 8.3
spxA-sg1	ATAATACTATATTGTTCGTC	49.4	25	9 ± 0.2
spxA-sg2	GTTATTTTGATTATCTGTCA	50.7	25	0.7 ± 0.02
hup1-sg1	AAACAAGCCTTAACAAGTTT	56.2	30	3 ± 0.1
hup1-sg2	ACTCTAAAAGTACCAAGTTC	54.5	35	22 ± 2.8
deoA-sg1	AAAGCGGCAGCTTGGTAGTC	65.6	55	11 ± 1.2
deoA-sg2	GCCTGAATGCATCATAGCGT	63.3	50	42 ± 8.5

## Data Availability

Data are contained in this manuscript or Appendix A. The datasets presented in this study can be found in online repositories. The names of the repository and accession number can be found in the article.

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
