# Peer review of "Gene Silencing through CRISPR Interference in Mycoplasmas"

_microorganisms, 2022, doi:10.3390/microorganisms10061159_

Round 1

Author Response

Dear reviewer, thank you for taking the time to assess our manuscript. We addressed all the concerns they you raised.

Comment 1:

Randomly inserting transgenes into an organism can be an easier route for transformation especially if there is a lack of tools for site specific insertions. However, it is crucial to understand that selecting transformants will lead to an experimental design where transformants must be investigated independently. The authors seem to understand that position effects matter on the expression levels of transgene but this is not reflected in their plots. It seems they pooled/lumped the expression levels of the transformants investigated into a bar plot. Unless there is a good reason to do this, I don’t think this how it should be done. Each transformant must be treated as independent and therefore the expression levels of their target genes should be compared amongst the other transformants obtained within the species. There will be differences and it is these difference that help describe the results. I believe that at minimum, the data points that represent the transformants should be indicated in the plots. I would rather the authors plot each transformant independently, though.

Response:  As reviewer noticed we understand that position of transgene can affect on its expression levels.  We really pooled the expression levels of dcas9 for several transformants on the bar plots and also expression level of target-genes in CRISPRi experiments. Standard deviation of our qRT-PCR data for different transformants did not exceed 15% in most of experiments. The strength of the promoter has a dominant effect on transgene expression. In order to demonstrate the expression levels of the transgene and target-genes in each individual transformants, we have updated Fig. 2, Fig. 4 and Fig. 5.

Comment 2:

I wonder why the authors generated transformants for mho, but do not fully characterize them like the mga transformants? Is there a reason for this? I was wondering why that data was excluded.

Response: We thank the reviewer for pointing this out. We performed gene-repression on Mycoplasma hominis (Mho) to demonstrate the universality of designed CRISPRi system. We showed the expression level of dcas9 was equal in both species and allowed successful repression of target-genes in Mga and Mho.

Comment 3: The authors performed some differential expression analyses (Fig 3), but do not indicate how they did this in the material and methods. Did they use RNA-Seq? If so, indicate this in the methods. The type of program and test that was used for the differential expression must be mentioned in the text. This lack of information is worrying.

Response: In our research we performed comparative proteomic analysis of two dcas9-expressing transformants and wild-type of Mycoplasma gallisepticum (Fig. 3B-C). It is not RNA-Seq. Liquid chromatography-mass spectrometry (LC-MS) was used for analysis of protein levels in our samples. We described statistical analysis in chapter “2.7 Proteomic analysis” in the section “2. Materials and Methods”. But in the original version of our manuscript, we did not describe in details the method of sample preparation for proteomic analysis (protein extraction, trypsinolysis and LC-MS analysis), referring to our previous article. This could have been a reason of misunderstanding for the reviewer. We have completed chapter “2.7 Proteomic analysis”, line 195-220.

Specific comments:

Line 52-53, Sentence does not go anywhere. Please elaborate.

Thank you, we have changed it, line 52-53

Line 59, Include the (S. pyogenes) after the first mention of Streptococcus pyogenes. Then use the short-hand throughout the text.

We modified this accordingly reviewer recommendation

Line 84, All protocols but the site directed mutagenesis is decribed. Please include description on how site-directed mutagenesis was performed. Include any relevant primers. Also, please start with a sentence

along the lines of, "To generate the XX plasmid(s) the following was done: First, the WT Cas9 was PCR amplified from S. pyogenes...."

We described more carefully how the site-directed mutagenesis was performed, line 90-106

Line 181-185, Raw files were extracted from what? Was the DNA sequence taken then translated to AA sequence?

Raw files (.raw) contain spectral data produced by Thermo Fischer Scientific's mass spectrometer. It is a standard format for proteomic data. This data was extracted from PRIDE database. It is our data that was previously deposited into database. We have completed chapter “2.7 Proteomic analysis”, line 195-235

Line 216-217, This should go in the methods.

We agree with the reviewer, we have changed the chapter "2.2 Plasmid construction" in the section "2. Materials and Methods". However, we have also kept this sentence in the chapter "3.1 Construction of a CRISPRi system for silencing gene expression in Mycoplasma". We believe that it is a reasonable.

Figure 1, Please clarify what this means and to which sequence this corresponds to. I'm unsure if it belongs to the top or bottom box.

We have changed Figure 1 and its legend. Please, check it out in the manuscript.

Line 221, I highly recommend using the circle tool if you are making this figure in powerpoint. The user-drawn lines do not help with the overall aesthetic of the article. If the circle tool cannot be used, I then recommend not drawing the circle part to indicate if the DNA is in a plasmid. I would indicate the plasmid name either top left of the box or top right.

Thank you, we have changed Figure 1.

Line221, Missing P in CRISPRi

We have fixed the error.

Line 221, Which panel in the figure does "A" correspond to? Please indicate on figure. The same goes for "B"

We have made the change.

Line 234-244, This belongs in the material and methods section.

Thank you for this suggestion. However, this part of manuscript contains how the pRLM5L2-dcas9 was derived. Probably, it partly duplicates the section "2. Material and Methods" but we believe it is necessary for better undersanding.

Line 246, I would remove these three words (The developed vector) from the title. We know the vector was developed.

We agree and have amended this sentence.

Line 246, This title is confusing. I would write something like, "....dCas9 is robustly expressed from the XXX plasmid in XX"

We have updated the title.

Line 248, Please indicate the time of colony appearance for the collections done. In addition, what was the author's criteria for choosing genomic insertions?

The first colonies of M. gallisepticum appeared on the fifth day of growth, colonies of M. hominis - on the eighth day. We have added this sentence in the chapter "2.3 Transformation", line 132-134. As we wrote, we selected colonies based on time of appearance and size. It was an indirect indicator of influence of transposon position to bacteria fitness. Then, for dcas9-expressing transformants, we mapped transposon insertion position into genome of M. gallisepticum and M. hominis through Sanger sequencing. The w3 primer was used (the location of this primer can be found in Fig. S1). This primer is located at a distance 218 nucleotides from one of the inverted repeats and is complementary to 5’-end of coding strand tetM. Two transformants with intergenic position of transposon were used for farther comparative proteomic analysis and growth rate.

Line 249-250, Please indicate what this is relative to (in terms of expression)

In original manuscript, line 249-250 – “The dCas9 expression was confirmed at both mRNA and protein levels using real-time quantitative PCR and Western blotting for three transformants of M. gallisepticum and two transformants for M. hominis (Fig. 2A,B).” Quantitative dcas9 expression was normalized to the expression level of tuf in qRT-PCR experiments for both mycoplasma species. So, it indicates that dcas9 was successfully transcribed to mRNA in two mycoplasma species (confirmed by qRT-PCR) and then this mRNA was translated by ribosomes to protein dCas9 (confirmed by Western-blot). Perhaps the using capital letter in dCas9 caused the question. We have changed dCas9 to dcas9.

Line 252, Panel C is not mentioned in text

Yes, it is our mistake. We have fixed it, line 293

Line 254, Please explain what the boxes indicate? Is the bold line in the middle a median or mean? Please explain.

In Figure 2A, boxplots represent a relative expression of dcas9 in M. gallisepticum and M. hominis. The relative expression was determined using the 2-ΔCt method and normalized to the amount of tuf transcripts present in the RNA samples. Bold line in boxplot is a median, upper line corresponds to the first quartile, lower line corresponds to the third quartile, whisker lines show the location of the minimum value on one side, and the maximum value on the other, individual point is an outlier. The mean value expression level of dcas9 was 0.43 for M. gallisepticum and 0.38 for M. hominis. Three independent transformants and three technical repeats for each was used.

Line 254-255, Shouldn't each transformant be plotted separately? The readers would want to see if there is some or no variation in expression amongst the transformants. Genomic position should affect this. Unless they are all inserted into the same site, please indicate this is the case.

We have taken reviewers advice into account and changed Figure 2, 4 and 5 to show expression level of dcas9 in individual transformants. The positions of transposon insertion into Mga and Mho genome were differed in all transformants. But the strength of promoter had a dominant effect on dcas9 expression and this level was equal in all selected transformants.

Line 269, Was there a reason the effect of dCas9 expression was not analyzed in Mho? Why was one species selected and not the other?

It would have been interesting to explore the effect of dcas9 expression on Mho. However, in the case of our study, it seems slightly out of scope because we have already described the effect of dcas9 on Mga and used Mho just demonstrate the universality of designed CRISPRi system. Mycoplasma hominis is an opportunistic urogenital pathogen in vertebrates. Among genital mycoplasmas, M. hominis is the most commonly reported to play a role in systemic infections and can persist in the host for a long time. Until recently, genetic manipulation for this specie was a challenge [Lartigue C & Bébéar C (2019) Random transposon insertion in the Mycoplasma hominis minimal genome. Sci Rep 9, 1–11.]. We believe that our CRISPRi system will help to discovery molecular mechanisms of persistence.

Line 278, remove “preliminarily” from sentence. Does not belong.

We have removed “preliminary”.

Line 280, explain “carrying capacity”

Carrying capacity is the maximum number, density, or biomass of a population that an environment can sustain. It is a common ecology definition [McArthur JV . (2006). Microbial Ecology: An Evolutionary Approach. Academic Press: Oxford, UK, pp. 136-139]. In our study, carrying capacity is a maximum value of growth index.

Line 334-336, It seems some repression was achieved. Unless you mean the the sgRNA against fur was not effective because it did not confer 100% repression? Please clarify. Unless the authors are referring to two different sgRNAs (same target sequence but different contexts). In either case, I am unsure what the results mean at this point

In original manuscript, the paragraph contained line 334-336 is about influence of terminator sequence of sgRNA on gene-repression. As we wrote, in initial studies, we used AT-rich terminator in sgRNA. The designed sgRNA1 containing AT-rich terminator for fur was not effective. We did not see any repression for fur (Fig. S3). But when we designed sgRNA1 containing canonical terminator from S. pyogenes, fur level was down in 22.5-fold. Both designed sgRNA1 contained the same 20-nt base-pairing region.

Line 340-343 and 346-350, this belongs in materials and methods

These sentences describe an important part of designed CRISPRi system and are discussed in the text. Removing them would complicate the understand of the manuscript.

Line 359-360, This sentence does not make sense.

In original manuscript, line 359-360 – “We generated sgRNAs (Table 1) that target the nontemplate strand of genes with the exact location dependent on the location of a protospacer adjacent motif (PAM) for dCas9 S. pyogenes”.  We have amended this sentence, line 401-402 – “We generated sgRNAs (Table 1) that target to the nontemplate strand of genes. The location of targets was adjacent to a protospacer motif (PAM) for dCas9 S. pyogenes.” In theory, sgRNAs can be designed to bind to either the template DNA strand or the nontemplate DNA strand. Binding specificity of sgRNA-dCas9 S. pyogenes complex is determined by both sgRNA-DNA base pairing and a short DNA motif (protospacer adjacent motif [PAM] sequence: NGG) juxtaposed to the DNA complementary region. In pioneer study of CRISPRi system, Lei S. Qi with colleagues demonstrated that only sgRNAs that bind to the nontemplate DNA strand showed silencing and PAM is required [Qi LS, Larson MH, Gilbert LA, Doudna JA, Weissman JS, Arkin AP & Lim WA (2013) Repurposing CRISPR as an RNA-guided platform for sequence-specific control of gene expression. Cell 152, 1173–1183.]. In our study, we designed sgRNAs that targets the nontemplate DNA strand of selected targets (Fur, WhiA, SpxA, Hup1 and DeoA). Moreover, all specific DNA targets were adjacent to PAM.

Line 363-364, Off target effects: Do you have a reference that can support this statement? If not, remove.

In original manuscript “Due to the high AT content of mycoplasmal genomes, the choice of the targeted region with downstream PAM 5'-NGG-3' is restricted. On the other hand, this factor can minimize off target effects.”

We do not have a reference; it is our assumption, but we can substantiate it. The dCas9 scans available DNA for a PAM before probing guide-target complementarity [Sternberg SH, Redding S, Jinek M, Greene EC, Doudna JA. DNA interrogation by the CRISPR RNA-guided endonuclease Cas9. Nature. 2014;507(7490):62-67. doi:10.1038/nature13011], [Anders C, Niewoehner O, Duerst A, Jinek M. Structural basis of PAM-dependent target DNA recognition by the Cas9 endonuclease. Nature. 2014;513(7519):569-573. doi:10.1038/nature13579].

So, we calculated the theoretical number of PAM sites NGG in coding sequences (CDS) of Mycoplasma gallisepticum, Escherichia coli and Bacillus subtilis. Given the total length of CDSs of these organisms, we calculated the number of PAM sites per 1000 bp. For AT-rich Mga this value was 22, for Escherichia coli with GC content 50.5%, the average number of PAM sites per 1000 bp was 56, for Bacillus subtilis – 47 (Table 1). It means that high GC content of genome correlates with higher frequency PAM motifs. Obviously, for successful sgRNA-dCas9 binding to DNA, strong base pairing between the 20 nt-region of sgRNA and the DNA target strand is necessary, but PAM serves as an essential gatekeeper preventing the dCas9 from accessing DNA sequences, even if they harbor complete complementarity to the 20 nt-region of sgRNA.

Table 1. The number of NGG in CDSs of three bacteria

Mga_S6

Eco_MG1655

Bacsu_substr168

Number of NGG in CDSs

19474

226325

174852

Total length of all CDSs, bp

885448

4025875

3731179

GC content, %

31

50.5

46.8

The average number of NGG in CDSs per 1000 bp

22

56

47

Line 365, “selected genes” Do you mean targeted genes?

Yes, reviewer is absolutely right. We have changed “selected genes” to “targeted genes”

Line 374-375, Why was deoA the only gene targeted in Mho?

It would have been interesting to repress more than one gene in Mho. In this study we have just demonstrated that our CRISPRi system can be used not only for Mga, but for Mho. Repressing one gene is enough for this goal. We show that we have a genetic tool for studying Mho and are going to use it in further study.

Line 375-376, Relevance of eno?

In original manuscript “The maximum value of repression was more than 40-fold with fur-sg2 in M. gallisepticum and deoA-sg2 in M. hominis. The expression level of eno encoding enolase was not changed for most transformants (Fig. S5).” We estimated the enolase expression to demonstrate that repression of target genes was not associated with a general drop of mRNA levels in transformants.

Line 392-394, Where are the stats for the nfo gene? Is it non significant? If so please indicate

We thank the reviewer for pointing this out. For nfo, the difference was not statistically significant. We have updated Figure 5A.

Line 394-396, Where are the transformants indicated on panel B? Again, where the results pooled? I recommend at least adding in the data points and indicating which transformants they correspond to. In panel B, I only see relative expression of the listed genes but I cannot compare them to the transformants. This plot needs to be redone.

We have added points to the plot. These points are corresponded the expression level of every gene in every transformant. In Figure 5B we compared the expression level of genes in transformants expressing dcas9 and the spxA-sgRNA1 (spxA-sg1) with dcas9-expressing transformants. In M. gallisepticum, 9-fold repression of spxA expression by CRISPR leads to 2.6-fold downregulation of osmC

Line 396-398, Why are we seeing eno expression here? Shouldn't we see whiA expression since rpsJ is relevant to whiA?

Expression level of eno was given to demonstrate that repression of whiA and activation of rpsJ was not associated with a general change of mRNA levels in transformants. We did not show expression level of whiA in Figure 5C, because it was already demonstrated in Figure 4.

Line 422-423, Indicate their importance..,

We have changed this sentence, line 469

Reviewer 2 Report

Please refer the attached file for comments.

Author Response

Dear reviewer, thank you for taking the time to assess our manuscript. We addressed all the concerns they you raised.

- Did the authors look for phenotypic effect if they are essential genes? I would recommend including growth data. These are simple experiments and will corroborate the presented data for gene repression.

We agree that it would be useful to demonstrate the phenotypic effect for transformants. We will take it in account in our further study. The data for gene repression was confirmed by qRT-PCR.

- Introduction does not cover the recent advancement in CRISPR studies in Mycoplasma. Here are some recent advancements to improve the introduction part that needs to be discussed and compared since it is in the same species doi: 10.1016/j.vetmic.2022.109436; doi: 10.1016/j.vetmic.2020.108868.

We agree and have updated, line 56-62

- There are some contradicting statements in introduction and results section that confuse the reader such as the author mention transform based vector are used for successful integration at random sites, but they have limitations. Then Why the authors used a transposon-based vector for establishing CRISPRi?

We have to use transposon-based vector for CRISPRi to design universal genetic tool, that can be used for different mycoplasma species. Unfortunately, there is no self-replicating plasmid, that can be stable in different mycoplasma species.

- Line 346 – 350 are confusing, it is not clear if the authors replaced promoter sequence in sg3 and sg4 to fine tune the expression or just stating a way to do so? If they are suggesting, then it should be part of discussion.

We thank the reviewer for pointing this out. We did not change promoter strength in this study because all constructed transformants were viable under this condition. But we have considered the possibility to do this if high expression level of sgRNA and as result a substantial repression of target gene would affect the bacterial survival.

- In Figure 4 out of 5 genes only spxA is the one where sgRNA2 has not given higher repression than sgRNA1. In this case both sgRNAs are targeting within 50 nt range from ATG. While in other cases sgRNA2 is targeting beyond 150 nt from start codon. Does the higher distance from the ATG generate stronger repression? Comment on it in discussion.

Thank you for pointing this out. In pioneer study of CRISPRi system, Lei S. Qi with colleagues demonstrated that silencing efficiency inversely correlated with the target distance from the translation start codon for E. coli [Qi LS, Larson MH, Gilbert LA, Doudna JA, Weissman JS, Arkin AP & Lim WA (2013) Repurposing CRISPR as an RNA-guided platform for sequence-specific control of gene expression. Cell 152, 1173–1183.]. In our study, when we chose the location of target sequences, we tried to use the nearest to start codon positions adjacent to PAM. For all target-genes except spxA, the sgRNA2 targeted further than sgRNA1 from the start codon was more efficient. This result differs from Qi 's conclusion. It is possible that the properties of sgRNAs have greater influence on the efficiency of gene repression in mycoplasmas. We did not find any dependence of sgRNAs activity on the GC content, but GU dinucleotide in the 3’ of targeting region of sgRNAs was associated with high repression of targets. There are seven theoretical suitable target sequences across spxA-coding DNA. The two selected spxA-sgRNAs did not have GU dinucleotide in the 3’. It would be very interesting to determine rules for selecting effective sgRNA for Mycoplasmas. We hope it will be our research topic. We have discussed it in the new version of manuscript, line 421-426

- I do not fully understand What was the purpose of using transposon-based vector since the CRISPRi based gene targeting fully relies on PAM site and 20 nucleotide base pairing sequences in sgRNA.

We used transposon-based vector to deliver dcas9 S. pyogenes and sgRNAs-coding sequences into mycoplasmas. Our transposon also contained tetM as a selective marker. This transposon is randomly integrated into genome after transformation. There is no universal self-replicating stable plasmid for mycoplasma species. You are absolutely right that CRISPRi based gene targeting fully relies on PAM site and 20 nucleotide base pairing sequences in sgRNA.

- It is not clear if RLM5L2-dCas9 is an integrative plasmid (role of transposase gene) or a replicative plasmid? The plasmid has a size of 64.5 kb. Is it normal for plasmids to be such large for Mycoplasma?

pRPLM5L2-dcas9 is an integrative Tn4001-based vector. It contains transposase-coding gene, tetM-coding gene (tetracycline resistance gene) and cloned dcas9-coding gene. These three genes are under mycoplasma promoters. Codon composition of tetM and transposase was optimized for expression in Mycoplasma. This plasmid is replicated in E. coli for enrichment. After transformation into mycoplasma, the region between inverted repeats (OIR and IIR) is randomly integrated into genome by Transposase. The pRLM5L2 has a size of 6.475 kbp.

- Figure 1 is not informative; label A and B are missing from Figure. Not clear which step is step 1 and which one is step 2. It is not clear when and How mutations were created to generate dCas9?

Thank you for pointing this out. We have changed Figure 1 and its legend. Please, check it out in the manuscript. We have also described more carefully how the site-directed mutagenesis of dcas9 was performed, line 90-106

- S. pyogenes dCas9 is easily available, why authors chose to go for a long error prone route with multiple PCR steps? Has the entire sequence verified by sequencing (including plasmid)?

Unfortunately, we had only wild-type cas9 S. pyogenes. Moreover, due to nonstandard genetic code in Mycoplasmas, we had to replace commonly used TGA stop codon which code Tryptophan in Mycoplasmas to TAA stop codon. To confirm that no errors were induced in PCR-amplified gene but only desired mutations, resulting pRLM5L2-dcas9 plasmid was a partly sequenced using Sanger method, line 105. We did not perform whole plasmid sequencing because we did not see the expediency in this.

- If I understand correctly TGA is a stop codon in Mycoplasma and TAA is also a stop codon? Why the authors change the TGA codon to TAA?

No, TGA (UGA) codes for tryptophan in Mycoplasmas [Yamao F, Muto A, Kawauchi Y, Iwami M, Iwagami S, Azumi Y, Osawa S. UGA is read as tryptophan in Mycoplasma capricolum. Proc Natl Acad Sci U S A. 1985] while in other bacteria it is a stop codon. We’ve changed our text to make it more clearly, line 252-254.

- Again, the generation of target specific sgRNA is a single PCR step method. But the authors used a combination of 7 oligonucleotides for creating every sgRNA. It is not clear to me What was the purpose behind using the seven oligonucleotides and how they were assembled? Refer this paper (doi: 10.1016/j.plasmid.2019.04.001) to include a pictorial representation of sgRNA cloning into dCas9 vector. It will be easier for reader to understand the assembly process.

Thank you very much for your suggestion. We are familiar with all the methods to generate target specific sgRNA. Our method seems to be easier and more adaptable for construction sgRNAs with different promoters. It does not require PCR. We just mixed seven oligonucleotides, phosphorylated them, added T4 DNA-ligase, incubated and cloned resulted product to pRLM5L2-dcas9 plasmid between ApaI and XbaI site. Purification after ligation of oligonucleotides is optional. All of these steps can be produced in one tube, all enzymes’ buffers are compatible. For changing targeted sequence, we just ordered three short oligonucleotide and generate other sgRNA-coding sequence. In further study we would like to design inducible promoter for sgRNA expression. We have changed Figure 1 and its legend to make it clear. The assembly of sgRNA-coding sequence is described in lines 107-117.

- The result section 3.4 does not provide and result and fits into M&M section.

 In original manuscript, the 3.4 result section is about influence of terminator sequence of sgRNA on gene-repression and also there is described an important part of designed CRISPRi system and are discussed in the text. Removing them would complicate the understand of the manuscript.

Round 2

Reviewer 1 Report

I accept the author's corrections.